# Tonic firing mode of midbrain dopamine neurons continuously tracks reward values changing moment-by-moment

Yawei Wang[1], Osamu Toyoshima[1], Jun Kunimatsu[1,2,3], Hiroshi Yamada[1,2,3], Masayuki Matsumoto[1,2,3]*

[1]Graduate School of Comprehensive Human Sciences, University of Tsukuba, Tsukuba, Japan; [2]Division of Biomedical Science, Faculty of Medicine, University of Tsukuba, Tsukuba, Japan; [3]Transborder Medical Research Center, University of Tsukuba, Tsukuba, Japan

**Abstract** Animal behavior is regulated based on the values of future rewards. The phasic activity of midbrain dopamine neurons signals these values. Because reward values often change over time, even on a subsecond-by-subsecond basis, appropriate behavioral regulation requires continuous value monitoring. However, the phasic dopamine activity, which is sporadic and has a short duration, likely fails continuous monitoring. Here, we demonstrate a tonic firing mode of dopamine neurons that effectively tracks changing reward values. We recorded dopamine neuron activity in monkeys during a Pavlovian procedure in which the value of a cued reward gradually increased or decreased. Dopamine neurons tonically increased and decreased their activity as the reward value changed. This tonic activity was evoked more strongly by non-burst spikes than burst spikes producing a conventional phasic activity. Our findings suggest that dopamine neurons change their firing mode to effectively signal reward values in a given situation.

*For correspondence:
mmatsumoto@md.tsukuba.ac.jp

Competing interests: The authors declare that no competing interests exist.

## Introduction

We often experience situations in which the value of future rewards changes on a moment-by-moment basis. For example, as a bartender pours a cocktail into your glass, the amount (i.e., the value) of the cocktail that you will receive gradually increases. Midbrain dopamine neurons have been considered to function as a key neural substrate that signals reward values and regulates reward-seeking behavior. These neurons respond to rewards and reward-predicting cues with phasic activity (*Bayer and Glimcher, 2005*; *Cohen et al., 2012*; *Kawagoe et al., 2004*; *Morris et al., 2004*; *Satoh et al., 2003*; *Schultz, 1998*). An influential theory proposed that such phasic activity of dopamine neurons represents reward prediction errors, discrepancies between obtained and predicted reward values, in reinforcement learning (*Doya, 2002*; *Montague et al., 1996*; *Schultz et al., 1997*), and the reward-evoked phasic activity of dopamine neurons has been shown to regulate this type of learning (*Chang et al., 2016*; *Stauffer et al., 2016*; *Steinberg et al., 2013*; *Tsai et al., 2009*). In addition, the cue-evoked phasic activity of dopamine neurons has been reported to reflect the value of cued rewards (*Matsumoto and Hikosaka, 2009*; *Roesch et al., 2007*; *Tobler et al., 2005*) and has recently been demonstrated to influence behavior associated with cued rewards (*Maes et al., 2020*; *Morrens et al., 2020*).

Although a wealth of studies have revealed roles of the phasic dopamine neuron activity in reward processing, it is unclear whether the same principle is applicable to real-life situations. Indeed, if reward values gradually change moment-by-moment as in the cocktail situation described above, the phasic activity of dopamine neurons, which is sporadic and has a short duration (around a couple of hundred milliseconds), likely fails to continuously track these changing values. Therefore,

the phasic activity seems to be unable to properly regulate reward-seeking behavior in changing environments in which reward values fluctuate. To address whether and how dopamine neurons signal changing reward values, we designed a Pavlovian procedure in which a visual cue indicated the value (i.e., the amount) of a liquid reward that gradually increased or decreased moment-by-moment. We recorded single-unit activity from dopamine neurons while macaque monkeys were conditioned with this Pavlovian procedure. Here, we demonstrate a tonic firing mode of dopamine neurons that effectively tracks changing reward values.

## Results

### Pavlovian procedure in which the value of a cued reward gradually changed

We designed a Pavlovian procedure with three different conditions (*Figure 1A–C*) and applied it to two macaque monkeys (monkey H and monkey P). In each condition, a bar stimulus with red and green areas was presented as a conditioned stimulus (CS) after a fixation period. The size of the green area indicated the amount (i.e., the value) of a liquid reward that the monkey would receive. The larger the green area became, the larger the associated reward value. In the first condition (value-increase condition, *Figure 1A*), the green area was minimal at the beginning and gradually increased (3.8°/s), that is, the reward value gradually increased (0.082 ml/s) (*Video 1*). The gradual increase randomly stopped within 2450 ms after the onset of the bar stimulus. In the second condition (value-decrease condition, *Figure 1B*), the green area was maximal at the beginning and gradually decreased (3.8°/s), that is, the reward value gradually decreased (0.082 ml/s) (*Video 2*). The gradual decrease randomly stopped within 2450 ms after the bar onset. In the third condition (value-fixed condition, *Figure 1C*), the size of the green area did not change and was instead fixed at the minimum, half, or maximum that predicted a 0.1, 0.2, or 0.3 ml reward, respectively. In addition to these CSs, a CS with a longitudinally halved green area was included and was randomly followed by a 0.1-0.3 ml reward (i.e., the reward value was uncertain). In each condition, the monkeys received the reward indicated by the green area immediately after the offset of the bar stimulus. The delay between the CS onset and the reward delivery was fixed at 2850 ms in each condition and did not change across trials. Each condition consisted of a block of 50 trials and was repeated once or more for each recording session (*Figure 1D*).

To examine whether the monkeys accurately predicted the reward values according to the size of the green area, we applied a choice task in which two bar stimuli with red and green areas were presented on the right and left sides of a fixation point (*Figure 1E*). The monkeys were required to choose one of the bar stimuli by making a saccade. These bar stimuli were identical to those used in the Pavlovian procedure except that the green area did not increase or decrease. The size of the green area indicated the reward value and was randomly assigned to each bar stimulus. We observed that the monkeys were more likely to choose the bar stimulus with the larger green area, that is, the logistic regression slope between the choice rate of the right bar stimulus and the value difference (right bar's value – left bar's value) was significantly larger than zero (monkey H: n = 85 sessions, mean ± SD = 0.94 ± 1.64, p=1.17 × $10^{-15}$; monkey P: n = 68 sessions, mean ± SD = 0.98 ± 1.65, p=7.64 × $10^{-13}$; Wilcoxon signed-rank test) (*Figure 1F*). This suggests that the monkeys correctly predicted the reward values according to the size of the green area.

### Tonic activity of dopamine neurons tracking gradually changing reward values

While the monkeys participated in the Pavlovian procedure, we recorded single-unit activity from 99 dopamine neurons (monkey H, n = 46; monkey P, n = 53) in and around the substantia nigra pars compacta (SNc) and the ventral tegmental area (VTA) (*Figure 2—figure supplement 1*, also see *Figure 2—figure supplement 2* for the characteristic spike waveforms of the recorded dopamine neurons). Here, we presumed that a possible way for dopamine neurons to signal changing reward values is to track the changing values by tonically increasing and decreasing their activity in parallel with the value change. Indeed, an example dopamine neuron tonically increased its activity as the reward value gradually increased in the value-increase condition (left in *Figure 2A*). This tonic increase in activity was clearly in contrast with the phasic responses to the CSs in the value-fixed

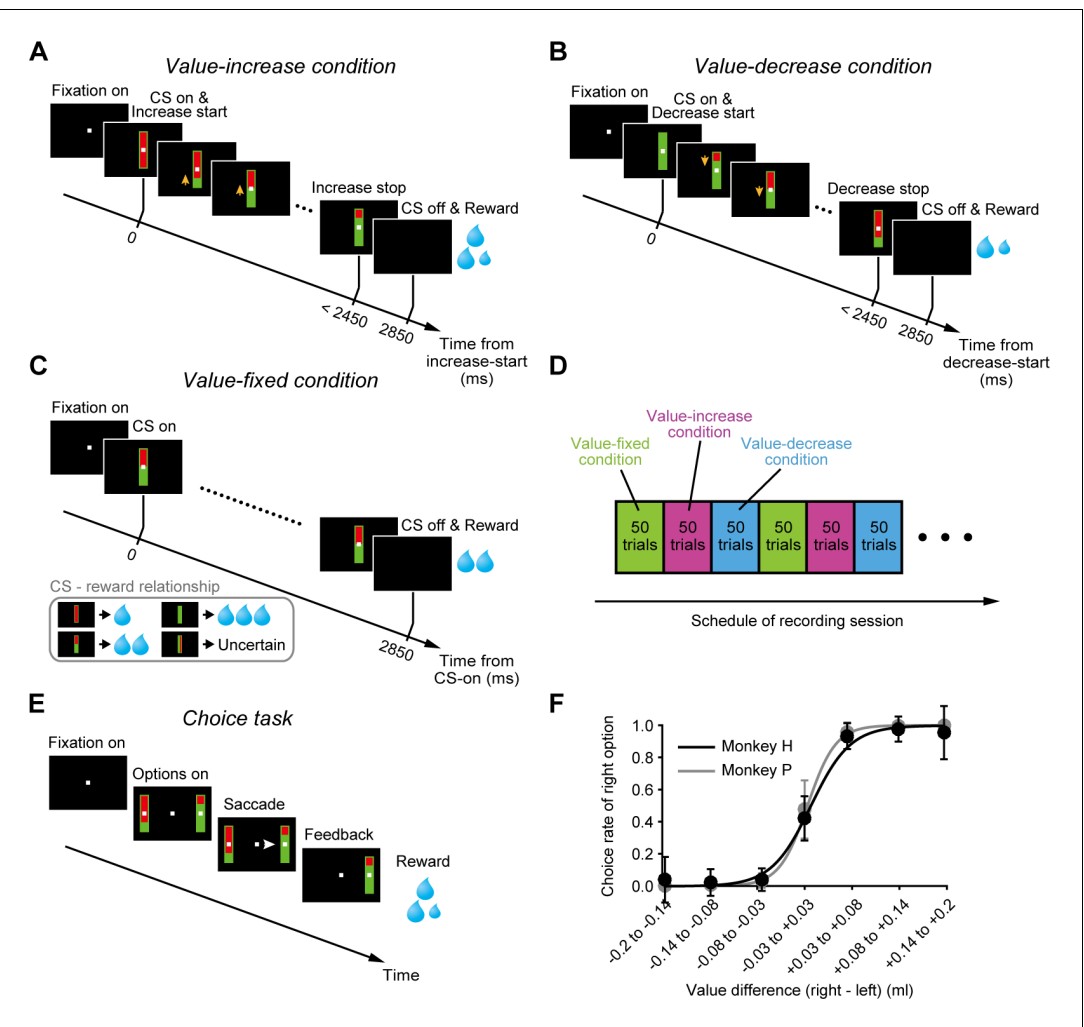

**Figure 1.** Pavlovian procedure in which the value of a cued reward gradually changed. (**A–C**) Value-increase (**A**), value-decrease (**B**), and value-fixed (**C**) conditions in the Pavlovian procedure. (**D**) Schedule of recording session. Each condition consisted of a block of 50 trials and was repeated once or more. (**E**) Choice task. (**F**) Choice rate of the right bar stimulus as a function of the value difference between the right and left bar stimuli in monkey H (n = 85 sessions; black) and monkey P (n = 68 sessions; gray). Error bars indicate SEM.

The online version of this article includes the following source data for figure 1:

**Source data 1.** Numerical data for *Figure 1F*.

condition, in which the reward value did not change (right in *Figure 2A*). This neuron did not exhibit clear activity modulation as the reward value gradually decreased in the value-decrease condition (middle in *Figure 2A*). Another example dopamine neuron tonically decreased its activity in the value-decrease condition (middle in *Figure 2B*), but it did not exhibit clear activity modulation in the value-increase condition (left in *Figure 2B*). This neuron also displayed phasic responses to the CSs in the value-fixed condition (right in *Figure 2B*).

To quantify tonic activity changes in the value-increase and value-decrease conditions, we calculated the slope of the regression line between firing rate and time for each neuron (thick gray lines in the left and middle columns of *Figure 2A,B*). In the value-increase condition, 19 of the 99 dopamine neurons exhibited a significantly positive slope (p<0.05; linear regression analysis), and, on average, the regression slope was significantly larger than zero (n = 99 neurons, mean ± SD = 0.046 ± 0.17, p=0.029; Wilcoxon signed-rank test) (*Figure 2C*). The averaged activity of the 19 dopamine neurons exhibited a clear tonic increase as the reward value gradually increased (left in *Figure 2E*) and displayed value-representing phasic responses to the CSs in the value-fixed

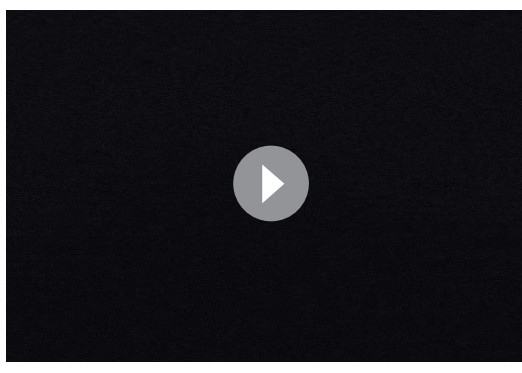

**Video 1.** Bar stimuli in the value-increase condition of the Pavlovian procedure. The size of the green area indicated the amount of a liquid reward. The green area was minimal at the beginning and gradually increased. The gradual increase randomly stopped within 2450 ms after the onset of the bar stimulus. https://elifesciences.org/articles/63166#video1

condition (significantly positive regression coefficient between firing rate and value: n = 19 neurons, mean ± SD = 1.61 ± 1.42, p=6.25 × 10$^{-4}$; Wilcoxon signed-rank test) (right in *Figure 2E*). In the value-decrease condition, 15 of the 99 dopamine neurons exhibited a significantly negative slope (p<0.05; linear regression analysis), and, on average, the regression slope was significantly smaller than zero (n = 99 neurons, mean ± SD = −0.048 ± 0.18, p=0.01; Wilcoxon signed-rank test) (*Figure 2D*). The averaged activity of the 15 dopamine neurons exhibited a clear tonic decrease as the reward value gradually decreased (left in *Figure 2F*) and displayed value-representing phasic responses to the CSs in the value-fixed condition (significantly positive regression coefficient between firing rate and value: n = 15 neurons, mean ± SD = 2.10 ± 0.97, p=6.10 × 10$^{-5}$; Wilcoxon signed-rank test) (right in *Figure 2F*). Notably, single dopamine neurons rarely exhibited both tonic increase and decrease in activity in the value-increase and value-decrease conditions, respectively (no significant correlation between the regression slopes in the value-increase and value-decrease conditions: n = 99 neurons, r = 0.048, p=0.64; correlation analysis) (*Figure 2—figure supplement 3*). These results suggest that a subset of dopamine neurons monitored the gradually increasing and decreasing reward values by tonically changing their activity, although the neurons did not signal these values in a unified manner.

To exclude the possibility that the tonic increase and decrease in activity were caused by spikes of non-dopamine neurons contaminating single-unit recording, we compared spike waveform (i.e., the width of spikes: a characteristic electrophysiological feature of dopamine neurons) between the time window during which the tonic activity changes occurred in the value-increase and value-decrease conditions and the time window during which the conventional, CS-aligned phasic dopamine responses occurred in the value-fixed condition (*Figure 2—figure supplement 4A,B*). There was no significant difference in spike width between the time windows for the 19 dopamine neurons with a tonic increase in activity in the value-increase condition (n = 19, spike width during the time window to detect the tonic activity increase in the value-increase condition: mean ± SD = 0.32 ± 0.08 ms, spike width during the time window to detect the CS-aligned phasic responses in the value-fixed condition: mean ± SD = 0.33 ± 0.10 ms, p=0.12; Wilcoxon signed-rank test) (*Figure 2—figure supplement 4A*) or for the 15 dopamine neurons with a tonic decrease in activity in the value-decrease condition (n = 15, spike width during the time window to detect the tonic activity decrease in the value-decrease condition: mean ± SD = 0.31 ± 0.07 ms, spike width during the time window to detect the CS-aligned phasic responses in the value-fixed condition: mean ± SD = 0.31 ± 0.06 ms, p=0.53; Wilcoxon signed-rank test) (*Figure 2—figure supplement 4B*). These results suggest that the tonic increase and decrease in activity were not caused by spikes of non-dopamine neurons contaminating single-unit recording. Furthermore, to exclude the possibility that the difference in activity patterns

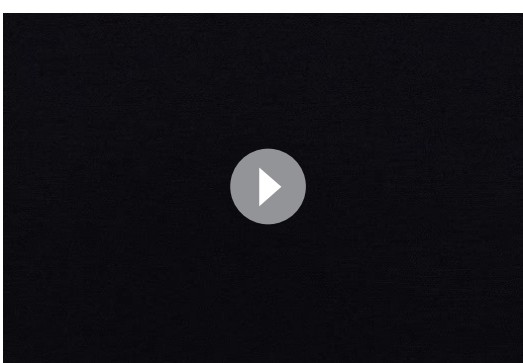

**Video 2.** Bar stimuli in the value-decrease condition of the Pavlovian procedure. The green area was maximal at the beginning and gradually decreased. The gradual decrease randomly stopped within 2450 ms after the onset of the bar stimulus. https://elifesciences.org/articles/63166#video2

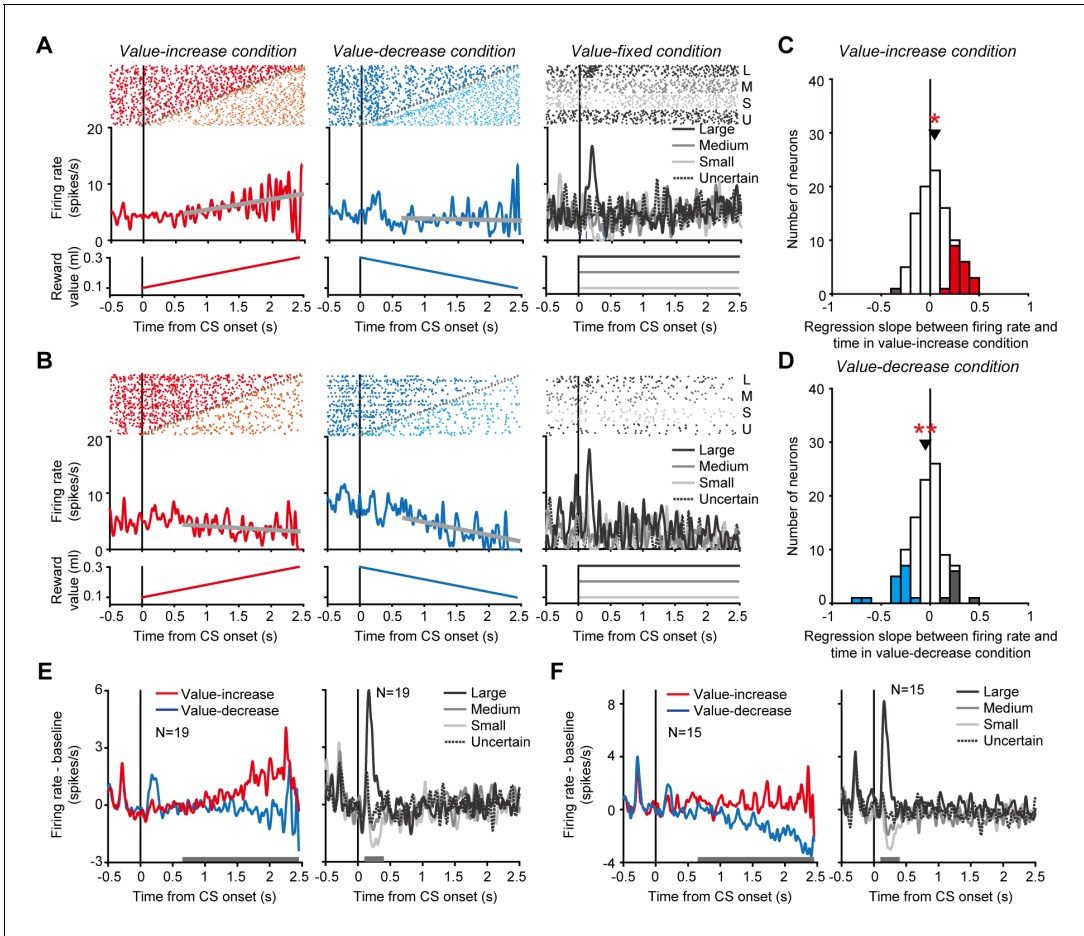

**Figure 2.** Tonic activity of dopamine neurons tracking gradually changing reward values. (A and B) Activity of two example dopamine neurons in the value-increase (left), value-decrease (middle), and value-fixed (right) conditions. Rasters and spike density functions (SDFs) are aligned by the conditioned stimulus (CS) onset. Left: Red and orange rasters indicate spikes occurring before and after, respectively, the reward value stopped increasing. Gray plots indicate the time at which the reward value stopped changing. The spikes occurring before the stop were used to calculate the SDF. The thick gray line indicates the regression line between firing rate and time. The bottom graph indicates the reward value as a function of time. Middle: Blue and cyan rasters indicate spikes occurring before and after, respectively, the reward value stopped decreasing. Other conventions are as in the left panel. Right: Black (top), dark gray, light gray, and black (bottom) rasters indicate spikes in trials in which the reward value was large, medium, small, and uncertain, respectively. Black (solid), dark gray, light gray, and black (dotted) curves indicate the SDFs in these trials. (C and D) Regression slope between firing rate and time in the value-increase (C) and value-decrease (D) conditions (n = 99 neurons). Red and gray bars indicate neurons with significantly positive and negative slopes, respectively, in the value-increase condition (C), while blue and gray bars indicate neurons with a significantly negative and positive slope, respectively, in the value-decrease condition (D) (p<0.05; linear regression analysis). Arrowheads indicate the mean regression slopes. Single and double asterisks indicate a significant deviation from zero (p<0.05 and <0.01, respectively; Wilcoxon signed-rank test). (E and F) Averaged SDFs of the 19 dopamine neurons with a significantly positive regression slope in the value-increase condition (E) and those of the 15 dopamine neurons with a significantly negative regression slope in the value-decrease condition (F). Left: Red and blue curves indicate the SDFs in the value-increase and value-decrease conditions, respectively. Right: Black (solid), dark gray, light gray, and black (dotted) curves indicate the SDFs in trials in which the reward value was large, medium, small, and uncertain, respectively. Horizontal gray bars indicate the time windows used for analyses.

The online version of this article includes the following source data and figure supplement(s) for figure 2:

**Source data 1.** Numerical data for *Figure 2A–F*.
**Figure supplement 1.** Localization of recording sites.
**Figure supplement 2.** Spike waveforms of dopamine neurons.
**Figure supplement 3.** Relationship between the regression slopes in the value-increase and value-decrease conditions.
**Figure supplement 4.** Waveforms of spikes causing the tonic activity changes and the conditioned stimulus (CS)-aligned phasic responses.
**Figure supplement 5.** Changes in the tonic activity of dopamine neurons between and within blocks.
**Figure supplement 6.** Effect of licking on dopamine neuron activity.
**Figure supplement 7.** Effect of eye movements on dopamine neuron activity.

among the value-increase, value-decrease, and value-fixed conditions (i.e., tonic increase, tonic decrease, and phasic changes, respectively) was due to the loss of spikes or emergence of spikes of other neurons during single-unit recording, we compared the baseline firing rate (i.e., firing rate from 500 to 0 ms before the fixation point onset) of the 19 and 15 dopamine neurons among the three conditions. On average, these was no significant difference in baseline firing rate among the conditions for the 19 dopamine neurons (n = 19, value-increase condition: mean ± SD = 3.4 ± 1.2 spikes/s, value-decrease condition: mean ± SD = 3.2 ± 1.3 spikes/s, value-fixed condition: mean ± SD = 3.6 ± 1.2 spikes/s, p=0.50; Kruskal-Wallis test) or the 15 dopamine neurons (n = 15, value-increase condition: mean ± SD = 3.7 ± 1.0 spikes/s, value-decrease condition: mean ± SD = 3.8 ± 1.3 spikes/s, value-fixed condition: mean ± SD = 3.8 ± 1.0 spikes/s, p=0.94; Kruskal-Wallis test). Even at the single-neuron level, only a few neurons (4 of the 19 neurons and 4 of the 15 neurons) exhibited a significant difference in baseline firing among the three conditions (p<0.05; Kruskal-Wallis test). These results suggest that the three different activity patterns were not due to the loss of spikes or emergence of other spikes during single-unit recording.

Although each condition consisted of a block of 50 trials and was repeated once or more for each recording session (*Figure 1D*), the tonic increase and decrease in activity were consistently observed between the first and latter blocks of each condition (*Figure 2—figure supplement 5A,B*). No neuron exhibited a significantly different regression slope between the first and latter blocks (p>0.05; comparison of two regression slopes). Furthermore, the tonic increase and decrease in activity were also consistently observed within each block (*Figure 2—figure supplement 5C,D*). No neuron exhibited a significantly different regression slope between the first half and latter half trials of blocks (p>0.05; comparison of two regression slopes). These results suggest that the tonic activity changes occurred in consistent manner between and within blocks.

A previous study reported that dopamine neurons exhibited a tonic increase in activity when future rewards were uncertain (*Fiorillo et al., 2003*). Thus, the tonic increase in activity, which we observed in the value-increase condition, might be accounted for by an uncertainty-evoked activity increase, because the reward value was uncertain until the green area stopped increasing in this condition. Contrary to this assumption, the 19 dopamine neurons, which exhibited a tonic increase in activity in the value-increase condition, did not exhibit a tonic increase in activity when the CS did not predict the future reward with certainty in the value-fixed condition (dotted curves in the right column of *Figure 2E*). On average, the regression slope between firing rate and time was not significantly different from zero (n = 19 neurons, mean ± SD = 0.015 ± 0.098, p=0.20; Wilcoxon signed-rank test). This result suggests that the tonic increase in activity in the value-increase condition was not evoked by uncertainty.

Recent studies using rodents have shown that a subgroup of dopamine neurons increases their activity when animals simply initiate a body movement (*da Silva et al., 2018*; *Howe and Dombeck, 2016*), although such movement-related dopamine neuron activation has not been reported in primates. In the present study, we observed that the monkeys sporadically licked the spout for reward delivery while the reward value gradually changed (see *Figure 2—figure supplement 6A and C* for the value-increase and value-decrease conditions, respectively). To examine whether the tonic increase and decrease in dopamine neuron activity were evoked by licking, we aligned the activity of the dopamine neurons exhibiting a tonic increase or decrease in activity by the onset of each licking event. These dopamine neurons did not exhibit clear activity modulation around the onset (see *Figure 2—figure supplement 6B* for the dopamine neurons with a tonic increase in activity in the value-increase condition, and see *Figure 2—figure supplement 6D* for the dopamine neurons with a tonic decrease in activity in the value-decrease condition), suggesting that the tonic increase and decrease in dopamine neuron activity were not caused by licking. We also observed that the monkeys often made eye movements along the vertical bar stimulus (i.e., CS) while the reward value gradually changed (see *Figure 2—figure supplement 7A and E* for example trials in the value-increase and value-decrease conditions, respectively). To examine whether the tonic increase and decrease in dopamine neuron activity were evoked by eye movements, we aligned the activities of the dopamine neurons exhibiting a tonic increase and decrease in activity by the onset of each saccade. These dopamine neurons did not exhibit clear activity modulation around the onset (see *Figure 2—figure supplement 7B* for the dopamine neurons with a tonic increase in activity in the value-increase condition, and see *Figure 2—figure supplement 7F* for the dopamine neurons with a tonic decrease in activity in the value-decrease condition). We further analyzed the relationship

between firing rate and vertical gaze position. In this analysis, we used the time window from 650 to 2450 ms (divided into 200 ms bins) after the CS onset (see the horizontal white bars in *Figure 2—figure supplement 7C,G*), which we used to calculate the regression slope between firing rate and time. In each 200 ms bin, we calculated the correlation coefficient between firing rate and vertical gaze position for each of the dopamine neurons with a tonic increase in activity in the value-increase condition (*Figure 2—figure supplement 7D*) and each of the dopamine neurons exhibiting a tonic decrease in activity in the value-decrease condition (*Figure 2—figure supplement 7H*). On average, these dopamine neurons did not exhibit a significant correlation coefficient between firing rate and vertical gaze position in either bin except the bin from 1650 to 1850 ms in the value-increase condition (n = 19 neurons, mean ± SD = −0.14 ± 0.20, p=0.011; Wilcoxon signed-rank test). These results suggest that the tonic increase and decrease in dopamine neuron activity were not caused by eye movements.

## Phasic activity of dopamine neurons evoked when the reward value stopped changing

We have observed the tonic activity of dopamine neurons that gradually increased and decreased as the reward value increased and decreased, respectively. We next found unique modulations of dopamine neuron activity that were evoked when the reward value stopped increasing and

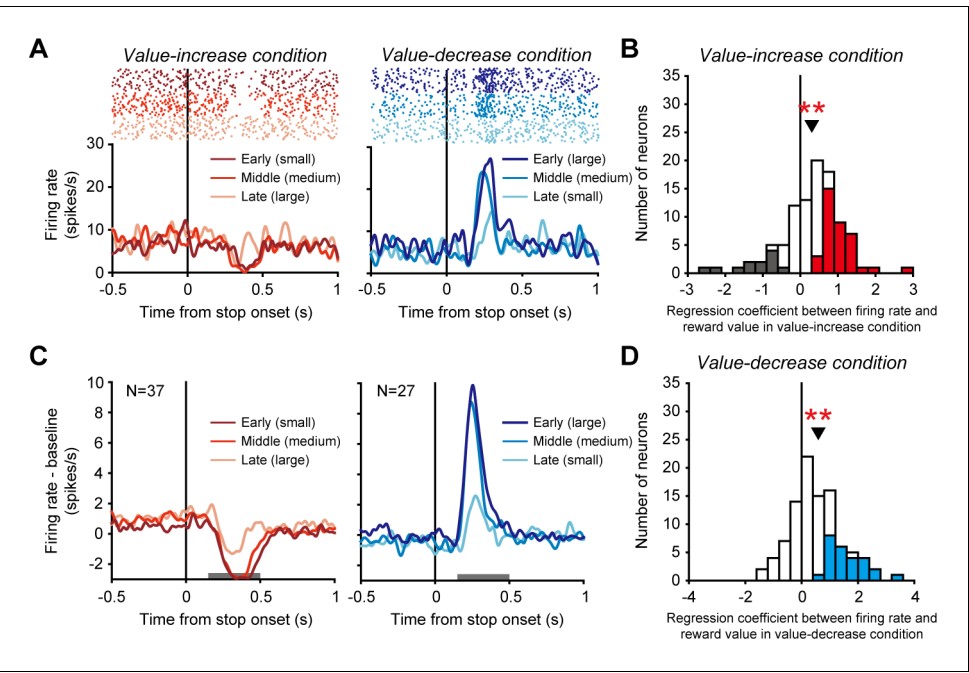

**Figure 3.** Phasic activity of dopamine neurons evoked when the reward value stopped changing. (**A**) Activity of an example dopamine neuron. Rasters and spike density functions (SDFs) are aligned by the time at which the reward value stopped increasing or decreasing in the value-increase (left) and value-decrease (right) conditions, respectively. Left: Dark red, red, and light red indicate the activity in trials in which the stopping of the value increase occurred at early (small reward 0.1-0.16 ml), middle (medium reward 0.16-0.23 ml), and late (large reward 0.23-0.3 ml) times, respectively. Right: Dark blue, blue, and light blue indicate the activity in trials in which the stopping of the value decrease occurred at early (large reward 0.23-0.3 ml), middle (medium reward 0.16-0.23 ml), and late (small reward 0.1-0.16 ml) times, respectively. (**B and D**) Regression coefficient between firing rate and reward value in the value-increase (**B**) and value-decrease (**D**) conditions (n = 99 neurons). Conventions are as in *Figure 2C,D*. (**C**) Averaged SDFs of the 37 dopamine neurons with a significantly positive regression coefficient in the value-increase condition (left) and those of the 27 dopamine neurons with a significantly positive regression coefficient in the value-decrease condition (right). Horizontal gray bars indicate the time windows used for analyses. Conventions are as in A.

The online version of this article includes the following source data for figure 3:

**Source data 1.** Numerical data for *Figure 3A–D*.

decreasing. An example dopamine neuron phasically decreased its activity when the value stopped increasing in the value-increase condition (left in *Figure 3A*) and phasically increased its activity when the value stopped decreasing in the value-decrease condition (right in *Figure 3A*). Such phasic decrease and increase in activity were generally observed across the recorded dopamine neurons and seem to be accounted for by conventional phasic dopamine inhibitory and excitatory responses to undesirable and desirable events. That is, the stopping of the value increase was undesirable and evoked the phasic decrease in dopamine neuron activity, whereas the stopping of the value decrease was desirable and evoked the phasic increase in dopamine neuron activity.

Notably, the earlier the reward value stopped increasing (i.e., the smaller the reward value), the more strongly the phasic activity decreased (significantly positive regression coefficient between firing rate and reward value: n = 99 neurons, mean ± SD = 0.31 ± 0.81, p=1.97 $\times$ 10$^{-5}$; Wilcoxon signed-rank test) (*Figure 3B*, also see left in *Figure 3C* for the averaged activity of 37 dopamine neurons with a significantly positive regression coefficient). Furthermore, the earlier the reward value stopped decreasing (i.e., the larger the reward value), the more strongly the phasic activity increased (significantly positive regression coefficient between firing rate and reward value: n = 99 neurons, mean ± SD = 0.59 ± 1.01, p=7.66 $\times$ 10$^{-8}$; Wilcoxon signed-rank test) (*Figure 3D*, also see right in *Figure 3C* for the averaged activity of 27 dopamine neurons with a significantly positive regression coefficient). Thus, the phasic decrease and increase in dopamine neuron activity likely reflected the reward value that was fixed when the green area stopped changing.

## Relationship between the tonic and phasic dopamine neuron activities

So far, we have reported that dopamine neurons exhibited not only the phasic activity but also the tonic one to effectively signal reward values. We next examined the relationship between the tonic and phasic activities. Even if dopamine neurons exhibited a cue-aligned phasic activity that represented the value of cued reward in the value-fixed condition, these neurons did not necessarily display a tonic increase in activity (i.e., a positive regression slope between firing rate and time) in the value-increase condition (no significant correlation between the regression coefficient and the regression slope: n = 99 neurons, r = −0.083, p=0.42; correlation analysis) (*Figure 4A*) or a tonic decrease in activity (i.e., a negative regression slope between firing rate and time) in the value-decrease condition (no significant correlation between the regression coefficient and the regression slope: n = 99 neurons, r = −0.003, p=0.97; correlation analysis) (*Figure 4B*).

Furthermore, even if dopamine neurons exhibited a stop-aligned phasic activity decrease that reflected the reward value in the value-increase condition, these neurons did not necessarily display a tonic decrease in activity (i.e., a negative regression slope between firing rate and time) in the value-decrease condition (no significant correlation between the regression coefficient and the regression slope: n = 99 neurons, r = −0.069, p=0.50; correlation analysis) (*Figure 4C*). Thus, there was no clear relationship between the phasic and tonic activity decreases. In addition, even if dopamine neurons exhibited a stop-aligned phasic activity increase that reflected the reward value in the value-decrease condition, these neurons did not necessarily display a tonic increase in activity (i.e., a positive regression slope between firing rate and time) in the value-increase condition (no significant correlation between the regression coefficient and the regression slope: n = 99 neurons, r = 0.061, p=0.55; correlation analysis) (*Figure 4D*). Thus, there was no clear relationship between the phasic and tonic activity increases. Taken together, the above results suggest that the tonic activity tracking the changing reward values was evoked independently of the phasic activity that represents the static, unchanged reward values.

We also examined the relationship between the cue-aligned phasic activity in the value-fixed condition and the stop-aligned phasic activity in the value-increase or value-decrease condition (*Figure 4E,F*). These phasic responses were significantly correlated with each other (the regression coefficient for the cue-aligned phasic activity in the value-fixed condition versus the regression coefficient for the stop-aligned phasic activity decrease in the value-increase condition: *Figure 4E*, n = 99 neurons, r = 0.48, p=5.6 $\times$ 10$^{-7}$; the regression coefficient for the cue-aligned phasic activity in the value-fixed condition versus the regression coefficient for the stop-aligned phasic activity increase in the value-decrease condition: *Figure 4F*, n = 99 neurons, r = 0.29, p=0.003; correlation analysis). Thus, the same dopamine neurons tended to exhibit both of the cue-aligned and stop-aligned phasic activities.

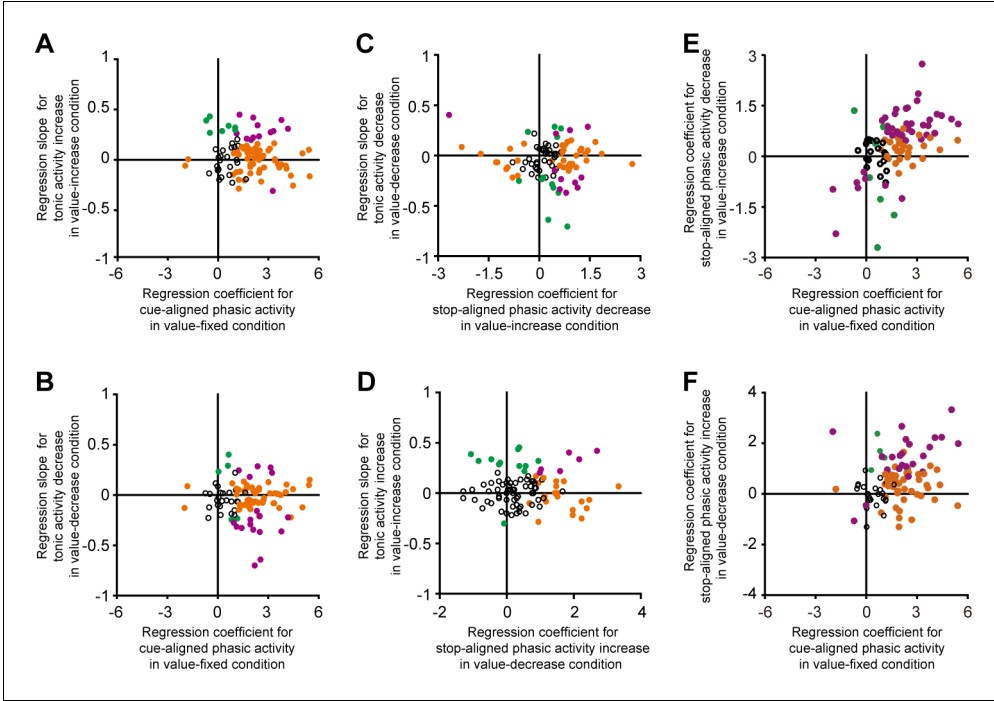

**Figure 4.** Relationship between the tonic and phasic dopamine neuron activities. (**A**) Regression slope between firing rate and time for the tonic activity increase in the value-increase condition (ordinate) and regression coefficient between firing rate and reward value for the cue-aligned phasic activity in the value-fixed condition (abscissa) (n = 99 neurons). (**B**) Regression slope between firing rate and time for the tonic activity decrease in the value-decrease condition (ordinate) and regression coefficient between firing rate and reward value for the cue-aligned phasic activity in the value-fixed condition (abscissa) (n = 99 neurons). (**C**) Regression slope between firing rate and time for the tonic activity decrease in the value-decrease condition (ordinate) and regression coefficient between firing rate and reward value for the stop-aligned phasic activity decrease in the value-increase condition (abscissa) (n = 99 neurons). (**D**) Regression slope between firing rate and time for the tonic activity increase in the value-increase condition (ordinate) and regression coefficient between firing rate and reward value for the stop-aligned phasic activity increase in the value-decrease condition (abscissa) (n = 99 neurons). (**E**) Regression slope between firing rate and reward value for the stop-aligned phasic activity decrease in the value-increase condition (ordinate) and regression coefficient between firing rate and reward value for the cue-aligned phasic activity in the value-fixed condition (abscissa) (n = 99 neurons). (**F**) Regression slope between firing rate and reward value for the stop-aligned phasic activity increase in the value-decrease condition (ordinate) and regression coefficient between firing rate and reward value for the cue-aligned phasic activity in the value-fixed condition (abscissa) (n = 99 neurons). Green and orange dots indicate neurons with a significant difference from zero in the ordinate and abscissa, respectively (p<0.05; linear regression analysis). Purple dots indicate neurons with significance for both. The online version of this article includes the following source data for figure 4:

**Source data 1.** Numerical data for *Figure 4A–F*.

## Burst and non-burst spike firing modes of dopamine neurons

Dopamine neurons are known to exhibit two different firing modes: burst spike firing and non-burst spike firing (*Grace and Bunney, 1984a*; *Grace and Bunney, 1984b*). The phasic activity increase of these neurons, which has been shown to reflect reward prediction errors, is produced by burst spike firing (*Bayer et al., 2007*). Here, we attempted to determine which firing mode produced the tonic activity increase of dopamine neurons. The burst and non-burst spike firing modes were identified based on interspike intervals (ISIs) using previously established criteria (*Grace and Bunney, 1984a*). These criteria were (1) an ISI of 80 ms or less to identify the onset of the burst spike firing mode and (2) an ISI of more than 160 ms to identify the end of the firing mode. We calculated the averaged activity of the 19 dopamine neurons that exhibited a tonic increase in activity in the value-increase condition separately for burst and non-burst spikes, and found that non-burst spike firing in these neurons increased tonically as the reward value gradually increased (left in *Figure 5A*). On the other

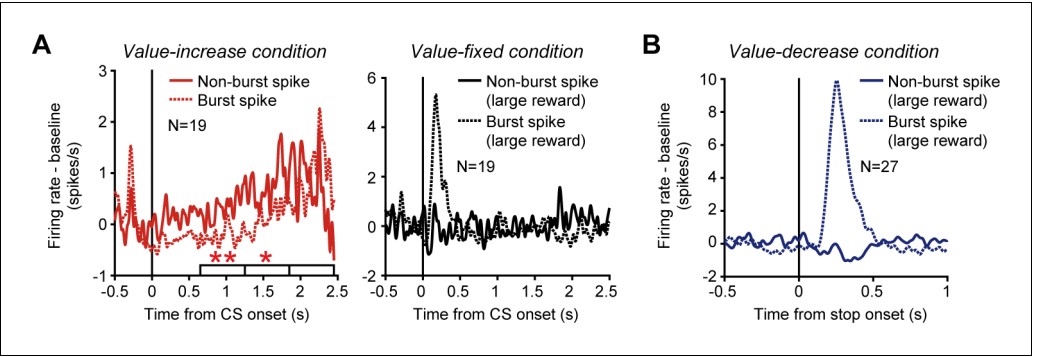

**Figure 5.** Phasic and tonic activities of dopamine neurons produced by burst and non-burst spikes. (**A**) Averaged spike density functions (SDFs) of the 19 dopamine neurons with a tonic increase in activity in the value-increase condition. Left: SDFs in the value-increase condition. Right: SDFs in the value-fixed condition. The SDFs are aligned by the conditioned stimulus (CS) onset and shown for burst (dotted curve) and non-burst (solid curve) spikes. Horizontal white bars indicate the initial, middle, and late periods to calculate the neural modulation between burst and non-burst spike firing. Single and double asterisks above the horizontal bars indicate periods during which the firing rate (original firing rate – baseline firing rate) was significantly larger for non-burst spike firing than for burst spike firing (p<0.05 and <0.01, respectively; bootstrap test with 1000 repetitions). The SDFs in the value-fixed condition were calculated using large reward trials. (**B**) Averaged SDFs of the 27 dopamine neurons with a phasic activity that was modulated by reward values when the values stopped decreasing in the value-decrease condition. The SDFs in large reward trials are shown and aligned by the time at which the reward value stopped decreasing. Other conventions are as in A.

The online version of this article includes the following source data for figure 5:

**Source data 1.** Numerical data for *Figure 5A,B*.

hand, burst spike firing in these neurons increased when the reward value came close to the maximum value near the end of the value-increase period. To statistically test this trend, we divided the calculation time window, which we used to detect the significantly positive regression slope of the 19 dopamine neurons, into initial, middle, and late periods (horizontal white bars in the left column of *Figure 5A*). We found that, while non-burst spike firing exhibited a significantly larger firing rate (original firing rate – baseline firing rate) compared with burst spike firing in the initial and middle periods, there was no significant difference in the late period (initial period: p=0.006; middle period: p=0.018; late period: p=0.43; bootstrap test with 1000 repetitions). Notably, these neurons exhibited a phasic increase in activity only with burst spike firing and not with non-burst spike firing in response to the CS predicting the large reward in the value-fixed condition (right in *Figure 5A*). In addition, the phasic increase in activity evoked when the reward value stopped decreasing in the value-decrease condition was also produced only by burst spike firing (*Figure 5B*). Taken together, these data indicate that, while the conventional phasic activity of dopamine neurons signaling the static, unchanged reward values was evoked by burst spike firing, the tonic activity tracking the changing reward values was produced more largely by non-burst spike firing.

## Discussion

Midbrain dopamine neurons respond to reward-predicting cues with phasic activity, and this phasic activity has been shown to represent the value of cued rewards (*Matsumoto and Hikosaka, 2009*; *Roesch et al., 2007*; *Tobler et al., 2005*). On the other hand, using a Pavlovian procedure in which a reward value gradually increased or decreased moment-by-moment, we found that a subset of dopamine neurons tracked the changing reward value by a tonic increase and decrease in activity rather than by the phasic activity. These neurons exhibited a conventional value-representing phasic activity when the reward value was static and when the value stopped changing. While the phasic activity was evoked by burst spike firing, the tonic activity tracking the changing reward value was produced not only by burst spike firing but also by non-burst spike firing. Particularly, the tonic activity was more largely produced by non-burst spike firing than burst spike firing.

## What variable does the tonic activity of dopamine neurons represent?

Although the phasic activity of dopamine neurons has attracted attention for its crucial roles in reward processing, recent studies using freely moving rodents reported that dopamine release in the striatum exhibited a tonic increase (or ramp) as animals approached rewards (*Engelhard et al., 2019*; *Hamid et al., 2016*; *Howe et al., 2013*; *Mohebi et al., 2019*). Such tonic increase has not been observed in dopamine neuron spike activity, at least in the VTA (*Mohebi et al., 2019*). Although the phasic activity of dopamine neurons has long been thought to represent reward prediction errors, the tonic increase in dopamine release has been proposed to represent 'state values' that gradually increase as animals spatially and temporally approach rewards (*Hamid et al., 2016*; *Mohebi et al., 2019*). However, it is not immediately clear whether the tonic increase and decrease in dopamine neuron spike activity, which we observed in head-fixed monkeys, represent state values or reward prediction errors. Because both the state values and reward prediction errors gradually changed in the same fashion in our Pavlovian procedure, we were unable to definitely identify which variable the tonic activity of dopamine neurons represented. On the other hand, we observed that, even if dopamine neurons exhibited a strong phasic activity, these neurons did not necessarily display a tonic increase or decrease in activity. That is, the tonic activity occurred independently of the phasic activity that has been thought to represent reward prediction errors. This suggests that the tonic activity might not represent reward prediction errors, and appears to be in accord with a recent perspective that dopamine neurons represent not only reward prediction errors but also multiple signals related to various events and behaviors, including punishments, movements, subjective sensory experience, response inhibition, decision-making, and working memory (*Brischoux et al., 2009*; *da Silva et al., 2018*; *de Lafuente and Romo, 2011*; *Engelhard et al., 2019*; *Howard et al., 2017*; *Howe and Dombeck, 2016*; *Matsumoto and Hikosaka, 2009*; *Matsumoto and Takada, 2013*; *Menegas et al., 2018*; *Ogasawara et al., 2018*; *Yun et al., 2020*).

An attribute that might cause the tonic increase and decrease in dopamine neuron activity is the motion of the green area. Since the green area dynamically increased and decreased in the value-increase and value-decrease conditions, respectively, in which we observed the tonic increase and decrease in dopamine neuron activity, it might be possible that these motions of the green area caused the tonic activity changes irrespective of the changing reward value. On the other hand, a previous study in monkeys observed that dopamine neurons exhibited a phasic activation but not tonic activity change in response to moving-dot images (*Nomoto et al., 2010*). According to this observation, the motion itself was unlikely to cause the tonic increase and decrease in dopamine neuron activity. Another attribute that might cause the tonic activity changes is the time distance to the reward delivery. This time distance became shorter as the green area increased and decreased. However, dopamine neurons exhibited the opposite modulations (i.e., the tonic increase and decrease in activity) in the value-increase and value-decrease conditions in which the time distance to the reward delivery equally became shorter as the green area changed. Therefore, the time distance was also unlikely to cause the tonic increase and decrease in dopamine neuron activity.

## What roles does the tonic activity of dopamine neurons play in behavior?

While the phasic dopamine activity, which is presumed to represent reward prediction errors, has been considered to provide teaching signals in reinforcement learning (*Doya, 2002*; *Montague et al., 1996*; *Schultz et al., 1997*), the tonic dopamine activity has been proposed to regulate motivation (*Cagniard et al., 2006*; *Niv, 2007*). A theory called 'incentive salience theory' proposes that dopamine signals assign incentive values to goals or actions of behavioral processes and motivate actions aimed at acquiring rewards (*Berridge and Robinson, 1998*). According to this theory, the tonic dopamine neuron activity, which we observed in the present study, seems to be a good candidate for a neural substrate that regulates the motivational vigor of actions in changing environments in which reward values fluctuate. That is, by tracking the changing values of future rewards, the tonic activity might enhance and suppress the motivational vigor of actions to obtain the rewards on a moment-by-moment basis.

## Conclusion

In conclusion, we demonstrated that, instead of phasic firing, tonic firing of dopamine neurons continuously tracked reward values changing moment-by-moment. This tonic activity was produced more largely by non-burst spikes than burst spikes that evoked the phasic dopamine neuron activity. Our findings expand the current knowledge on dopamine neuron signals by highlighting the alternating firing modes of these neurons, which effectively signal reward values in a given situation. Further investigations are called for to definitely determine whether the tonic activity of dopamine neurons represents state values, reward prediction errors, or other reward-related variables.

## Materials and methods

### Key resources table

| Reagent type (species) or resource | Designation | Source or reference | Identifiers | Additional information |
|---|---|---|---|---|
| Biological sample (*Macaca fuscata*) | *Macaca fuscata* | National Bio Resource Project of the MEXT, Japan | | |
| Software, algorithm | MATLAB | MathWorks | RRID:SCR_001622 | |

### Animals

Two adult macaque monkeys (*Macaca fuscata*; monkey H, female, 6.6 kg, 8 years old; monkey P, female, 6.8 kg, 7 years old) were used for the experiments. All procedures for animal care and experimentation were approved by the University of Tsukuba Animal Experiment Committee (permission number 14–137), and were carried out in accordance with the guidelines described in *Guide for the Care and Use of Laboratory Animals* published by the Institute for Laboratory Animal Research.

### Behavioral tasks

Behavioral tasks were controlled using MATLAB (Mathworks, MA) with Psychtoolbox, a freely available toolbox for MATLAB. The monkeys sat in a primate chair facing a computer monitor in a sound-attenuated and electrically shielded room. Eye movements were monitored using an infrared eye-tracking system (EYE-TRAC 6; Applied Science Laboratories, MA) with sampling at 240 Hz. A liquid reward was delivered through a spout that was positioned in front of the monkey's mouth. Licking of the monkeys was monitored during the recording of 68 of the 99 recorded dopamine neurons. To monitor licking, a strain gauge was attached to the spout and measured the strain of the spout caused by licking.

We designed a Pavlovian procedure with three different conditions (*Figure 1A–C*) and applied it to the two monkeys. In each condition, trials started with the presentation of a central fixation point (0.8°×0.8°). The monkey was required to fixate this point. After a 400 ms fixation period, a bar stimulus (2.9°×9.4°) with red and green areas was presented as a CS. The size of the green area indicated the amount (i.e., the value) of a liquid reward that the monkey would receive. The larger the green area became, the larger the associated reward value. In the first condition (value-increase condition, *Figure 1A*), the green area was minimal at the beginning and gradually increased (3.8°/s), that is, the reward value gradually increased (0.082 ml/s) (*Video 1*). The gradual increase randomly stopped within 2450 ms after the onset of the bar stimulus (uniform distribution from 0 to 2450 ms) so that the monkey was unable to precisely predict the reward value until the green area stopped increasing. In the second condition (value-decrease condition, *Figure 1B*), the green area was maximal at the beginning and gradually decreased (3.8°/s), that is, the reward value gradually decreased (0.082 ml/s) (*Video 2*). The gradual decrease randomly stopped within 2450 ms after the bar onset (uniform distribution from 0 to 2450 ms). In the third condition (value-fixed condition, *Figure 1C*), the size of the green area did not change and was instead fixed at the minimum, half, or maximum that predicted a 0.1, 0.2, or 0.3 ml reward, respectively (these reward amounts were the same as those indicated by the green area in the value-increase and value-decrease conditions). In addition to these CSs, a CS constituting of vertical red and green bars was included, and this was randomly followed by a 0.1-0.3 ml reward (i.e., the reward value was uncertain). These four CSs were presented with an equal probability (25% each). In each condition, the total time during which the CS was presented

was fixed at 2850 ms, and the monkey was required to maintain the central fixation during this period. Immediately after the offset of the bar stimulus, the reward indicated by the green area was delivered simultaneously with a tone (1000 Hz). The delay between the CS onset and the reward delivery did not change across trials. Each condition consisted of a block of 50 trials, and we collected data by repeating the three conditions once or more for each neuron (*Figure 1D*). The order of the three conditions, (1) the value-fixed, (2) value-increase, and (3) value-decrease conditions, was fixed across recording sessions. The total amount of reward was the same (10 ml) among blocks. Trials were aborted immediately if the monkey (1) did not start the central fixation within 4000 ms after the onset of the fixation point or (2) broke the central fixation during the initial 400 ms fixation period or the 2850 ms CS period (i.e., a continuous 3250 ms fixation was required). These error trials were signaled by a beep tone (100 Hz) and excluded from analyses.

We also used a choice task (*Figure 1E*). Each trial started with the presentation of a central fixation point (0.8°×0.8°), and the monkey was required to fixate this point. After a 400 ms fixation period, two bar stimuli with red and green areas (2.9°×9.4°) were presented on the right and left sides of the fixation point (eccentricity: 8.8°). The monkey was required to choose one of the bar stimuli by making a saccade immediately after the presentation of the bar stimuli. These bar stimuli were identical to those used in the Pavlovian procedure except that the green area did not increase or decrease over time. The size of the green area indicated the value of the reward that the monkey would obtain by choosing that bar stimulus and was randomly assigned to each bar stimulus (uniform distribution from 0.1 to 0.3 ml). Immediately after the monkey chose a bar stimulus, the other bar stimulus disappeared. Then, the reward indicated by the green area of the chosen bar stimulus was delivered simultaneously with a tone (1000 Hz). Trials were aborted immediately if the monkey (1) did not start the central fixation within 4000 ms after the onset of the fixation point or (2) broke the central fixation during the 400 ms fixation period. These error trials were signaled by a beep tone (100 Hz) and excluded from analyses.

## Single-unit recording

A plastic head holder and recording chamber were fixed to the skull under general anesthesia and sterile surgical conditions. The recording chamber was placed over the midline of the frontoparietal lobes, and aimed at the SNc and VTA in both hemispheres. The head holder and recording chamber were embedded in dental acrylic that covered the top of the skull and were firmly anchored to the skull by plastic screws.

Single-unit recordings were performed using tungsten electrodes with impedances of approximately 3.0 MΩ (Frederick Haer, ME) that were introduced into the brain through a stainless steel guide tube using an oil-driven micromanipulator (MO-97-S; Narishige, Tokyo, Japan). Recording sites were determined using a grid system, which allowed recordings at every 1 mm between penetrations. For a finer mapping of neurons, we also used a complementary grid that allowed electrode penetrations between the holes of the original grid.

Electrophysiological signals were amplified, band-pass filtered (200 Hz to 3 kHz; RZ5D, Tucker-Davis Technologies, FL), and stored in a computer at the sampling rate of 24.4 kHz. Single-unit potentials were isolated using a window discrimination software (OpenEx, Tucker-Davis Technologies, FL).

## Localization and identification of dopamine neurons

We recorded single-unit activity from putative dopamine neurons in the SNc and VTA. To localize the recording regions, the monkeys underwent an MRI scan to determine the positions of the SNc and VTA (*Figure 2—figure supplement 1*). Putative dopamine neurons were identified based on their well-established electrophysiological signatures: a low background firing rate at around 5 Hz, a broad spike potential in clear contrast to neighboring neurons with a high background firing rate in the substantia nigra pars reticulata (*Figure 2—figure supplement 2*), and a phasic excitation in response to free reward.

## Data analysis

For null hypothesis testing, 95% confidence intervals (p<0.05) were used to define statistical significance in all analyses.

To examine whether the monkeys accurately predicted the reward values according to the size of the green area (*Figure 1F*), the choice rate of the right bar stimulus was fitted by the following logistic function:

$$P = \frac{1}{1 + \exp(-(\beta_0 + \beta_1 \times (V_{right} - V_{left})))}$$

where $P$ indicates the choice rate of the right bar stimulus, $V_{right}$ and $V_{left}$ indicate the reward values obtained by choosing the right and left bar stimuli, respectively, and $\beta_0$ and $\beta_1$ indicate the coefficients determined by logistic regression.

To calculate spike density functions (SDFs), each spike was replaced by a Gaussian curve ($\sigma$ = 15 ms).

To calculate SDFs aligned by the CS onset in the value-increase and value-decrease conditions (*Figure 2A,B,E,F* and *Figure 5A*), spikes that occurred before the reward value stopped increasing or decreasing were used.

To calculate averaged SDFs across neurons (*Figure 2E,F*, *Figure 3C*, and *Figure 5A,B*), the baseline firing rate of each neuron was measured using a window from 500 to 0 ms before the fixation point onset and was subtracted from the original firing rate of each neuron.

To quantify tonic activity changes in the value-increase and value-decrease conditions (*Figure 2C, D*), the slope of the regression line between firing rate and time was calculated for each neuron. First, a window from 650 to 2450 ms after the CS onset was divided into 200 ms bins, and the firing rate in each bin was measured. Then, the regression line between the firing rate in each bin and the time at the center of each bin was calculated. The calculation window from 650 to 2450 ms after the CS onset was determined to exclude the effect of phasic responses evoked immediately after the CS onset.

To examine phasic activity modulations evoked by the reward value in the value-fixed condition, the regression coefficient between firing rate and reward value was calculated for each neuron. The firing rate was measured using a window from 100 to 400 ms after the CS onset. The calculation window was determined such that the window includes the major part of the neural modulation in the averaged activity.

To calculate SDFs aligned by the onset of each saccade (*Figure 2—figure supplement 7B,F*), the onset was determined as the time when angular eye velocity exceeded 40°/s.

To examine phasic activity modulation evoked when the reward value stopped increasing or decreasing in the value-increase and value-decrease conditions (*Figure 3B,D*), the regression coefficient between firing rate and reward value was calculated for each neuron. First, all trials were divided into three groups based on the reward value (large 0.23-0.3 ml, medium 0.16-0.23 ml, small 0.1-0.16 ml), and the firing rate was measured using a window from 150 to 500 ms after the stop onset for each trial group. Then, the regression coefficient between the firing rate and the mean reward value in each trial group was calculated. The calculation window was determined such that the window includes the major part of the neural modulation in the averaged activity.

To statistically test the difference in firing rate (original firing rate – baseline firing rate) between burst and non-burst spike firing (left in *Figure 5A*), a bootstrap procedure was applied. We first divided the calculation time window (650-2450 ms after the CS onset), which we used to detect the significantly positive regression slope of the 19 dopamine neurons (horizontal gray bar in the left column of *Figure 2E*), into initial (650-1250 ms), middle (1250-1850 ms), and late (1850-2450 ms) periods. The 19 dopamine neurons were randomly resampled with replacements to form a new bootstrap dataset that had the same number of neurons as the original dataset. Using the new dataset, we compared the firing rate (original firing rate – baseline firing rate) between burst and non-burst spike firing for the initial, middle, and late periods. This random resampling and comparison process was repeated 1000 times. If the firing rate was larger for burst spike firing than for non-burst spike firing or vice versa in more than 975 repetitions, the difference in firing rate was regarded as significant (p<0.05; bootstrap test with 1000 repetitions).

## Acknowledgements

We thank K Bunzui, T Okano, and Y Narita for animal care, and Dr. T Kawai for supporting analyses. This research was supported by MEXT KAKENHI Grant Number JP16H06567 to MM, and JST CREST

Grant Number JPMJCR1853 to MM. The monkeys used in the present study were provided by NBRP 'Japanese Monkeys' through the National Bio Resource Project of the MEXT, Japan.

## Additional information

### Funding

| Funder | Grant reference number | Author |
|---|---|---|
| Ministry of Education, Culture, Sports, Science and Technology | KAKENHI JP16H06567 | Masayuki Matsumoto |
| Japan Science and Technology Agency | CREST JPMJCR1853 | Masayuki Matsumoto |

The funders had no role in study design, data collection and interpretation, or the decision to submit the work for publication.

### Author contributions

Yawei Wang, Osamu Toyoshima, Data curation, Formal analysis, Investigation, Methodology, Writing - review and editing; Jun Kunimatsu, Hiroshi Yamada, Investigation, Methodology, Writing - review and editing; Masayuki Matsumoto, Conceptualization, Supervision, Funding acquisition, Investigation, Methodology, Writing - original draft, Project administration, Writing - review and editing

### Author ORCIDs

Yawei Wang (iD) http://orcid.org/0000-0003-3437-6658
Osamu Toyoshima (iD) https://orcid.org/0000-0003-1905-8483
Jun Kunimatsu (iD) https://orcid.org/0000-0002-8003-0650
Hiroshi Yamada (iD) http://orcid.org/0000-0001-8155-8847
Masayuki Matsumoto (iD) https://orcid.org/0000-0003-0832-1565

### Ethics

Animal experimentation: All procedures for animal care and experimentation were approved by the University of Tsukuba Animal Experiment Committee (permission number, 14-137), and were carried out in accordance with the guidelines described in Guide for the Care and Use of Laboratory Animals published by the Institute for Laboratory Animal Research.

### Decision letter and Author response

Decision letter https://doi.org/10.7554/eLife.63166.sa1
Author response https://doi.org/10.7554/eLife.63166.sa2

## Additional files

### Supplementary files

• Transparent reporting form

### Data availability

Source data files have been provided for all figures.

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
