## [Decision Letter]

**Acceptance summary:**

This study examined how the activity of dopamine neurons change when the expected amount of reward changes gradually and continuously, rather than abruptly as when a conditioned stimulus is suddenly presented. Although the activity of some dopamine neurons gradually increased or decreased as the expected reward changes, this was not correlated with the phasic signals of dopamine neurons, suggesting that the phasic and tonic reward prediction error signals of dopamine might be processed by separate channels. The experiments are designed cleverly and the analysis is performed well. This work will be of interest to those interested in dopamine function, reinforcement learning, and/or decision making.

**Decision letter after peer review:**

Thank you for submitting your article "Tonic firing mode of midbrain dopamine neurons continuously tracks reward values changing moment-by-moment" for consideration by *eLife*. Your article has been reviewed by two peer reviewers, including Daeyeol Lee as the Reviewing Editor and Reviewer #1, and the evaluation has been overseen by Kate Wassum as the Senior Editor. The following individual involved in review of your submission has agreed to reveal their identity: Matthew Roesch (Reviewer #2).

The reviewers have discussed the reviews with one another and the Reviewing Editor has drafted this decision to help you prepare a revised submission.

Summary:

The authors investigated whether and how the activity of dopamine neurons changes when the expected amount of reward changes gradually and continuously, rather than abruptly as when a conditioned stimulus is suddenly presented. This is an important question, since the commonly accepted reward prediction error RPE hypothesis would predict that the same dopamine neurons encoding the RPE would also gradually increase or decrease their activity when the expected reward changes gradually. Although the authors found such signals, interestingly, they were not correlated with the phasic signals of dopamine neurons, suggesting that the phasic and tonic RPE signals of dopamine might be processed by separate channels. The experiments are designed cleverly and the analysis is performed well. Nevertheless, there are some concerns that must be addressed.

Essential revisions:

1) Subsection “Tonic activity of dopamine neurons tracking gradually changing reward values”. The authors state that in the value-fixed condition, dopamine neurons did not exhibit a tonic increase, but do not provide any statistical results. Please include these.

2) In this experiment, dopamine neurons can display two different phasic signals, one associated with the CS presentation, and the other associated with the end of value increase or decrease. One question that was not addressed is whether the same dopamine neurons tended to display both of these phasic signals. The authors should include this information.

3) The authors should address the possibility of whether neurons with tonic modulation might not be DA. For example, were waveforms that had tonic changes during trials any different than waveforms that had phasic changes?

4) Several previous studies have suggested that DA encodes movement parameters. Therefore, it would be important for the authors to exclude the following confounding factors.

a) Gaze (smooth pursuit tracking the stimulus). Was activity correlated to gaze?

b) Licking. Did monkeys' ramp up and decrease anticipatory licking during these tonic changes in firing? Was activity correlated to licking?

5) Did the total amount of reward differ between blocks of trials? If so, was this reflected in the firing?

6) Did tonic firing patterns change across blocks of trials and were they consistent between the first and second runs of the same trial block?

7) Since this was a blocked paradigm, some of the effect might be due to drift in the signal or loss of cells or emergence of new cells. How stable were the recording over time?

8) The authors might be going one step too far by suggesting this might be involved in action selection. Why didn't they record during the choice task?

---

## [Author Response]

Essential revisions:1) Subsection “Tonic activity of dopamine neurons tracking gradually changing reward values”. The authors state that in the value-fixed condition, dopamine neurons did not exhibit a tonic increase, but do not provide any statistical results. Please include these.

Thank you very much for this critical comment. We added statistical results in the revised manuscript as follows:

“Contrary to this assumption, the 19 dopamine neurons, which exhibited a tonic increase in activity in the value-increase condition, did not exhibit a tonic increase in activity when the CS did not predict the future reward with certainty in the value-fixed condition (dotted curves in the right column of Figure 2E). On average, the regression slope between firing rate and time was not significantly different from zero (n = 19 neurons, mean ± SD = 0.015 ± 0.098, p = 0.20; Wilcoxon signed-rank test).”

2) In this experiment, dopamine neurons can display two different phasic signals, one associated with the CS presentation, and the other associated with the end of value increase or decrease. One question that was not addressed is whether the same dopamine neurons tended to display both of these phasic signals. The authors should include this information.

The information pointed out above is very important for readers to understand the relationship between the phasic response evoked by the CS presentation in the value-fixed condition and the phasic response evoked at the end of value increase or decrease, because we postulate that both of these phasic responses encode the same conventional signal, i.e., reward prediction error. To investigate whether the same dopamine neurons tended to display both of the phasic responses, we calculated the correlation between the magnitudes of these responses (new Figure 4E and F in the revised manuscript). We found that these phasic responses significantly correlated with each other, suggesting that the same dopamine neurons tended to display both of the phasic responses. We added this result in the revised manuscript as follows:

“We also examined the relationship between the cue-aligned phasic activity in the value-fixed condition and the stop-aligned phasic activity in the value-increase or value-decrease condition (Figure 4E, F). […] Thus, the same dopamine neurons tended to exhibit both of the cue-aligned and stop-aligned phasic activities.”

3) The authors should address the possibility of whether neurons with tonic modulation might not be DA. For example, were waveforms that had tonic changes during trials any different than waveforms that had phasic changes?

Thank you very much for this critical comment. To exclude the possibility that the tonic increase and decrease in activity were caused by spikes of non-dopamine neurons contaminating single-unit recording, we compared spike waveforms that caused these tonic activity changes and spike waveforms that caused the conventional, CS-aligned phasic dopamine responses in the control (i.e., value-fixed) condition. Consequently, we observed that these spike waveforms (i.e., the widths of these spikes) were not significantly different. This result suggests that the tonic increase and decrease in activity were not caused by spikes of non-dopamine neurons contaminating single-unit recording. We added the description of these analysis and result in the revised manuscript as follows:

“To exclude the possibility that the tonic increase and decrease in activity were caused by spikes of non-dopamine neurons contaminating single-unit recording, we compared spike waveform (i.e., the width of spikes: a characteristic electrophysiological feature of dopamine neurons) between the time window during which the tonic activity changes occurred in the value-increase and value-decrease conditions and the time window during which the conventional, CS-aligned phasic dopamine responses occurred in the value-fixed condition (Figure 2—figure supplement 4A, B). […] These results suggest that the tonic increase and decrease in activity were not caused by spikes of non-dopamine neurons contaminating single-unit recording.”

4) Several previous studies have suggested that DA encodes movement parameters. Therefore, it would be important for the authors to exclude the following confounding factors.a) Gaze (smooth pursuit tracking the stimulus). Was activity correlated to gaze?b) Licking. Did monkeys' ramp up and decrease anticipatory licking during these tonic changes in firing? Was activity correlated to licking?

This is a really important point. The reviewer raised the possibility that the monkeys increased and decreased anticipatory licking as the reward value gradually increased and decreased, respectively. The tonic increase and decrease in dopamine neuron activity might be caused by such liking rather than the gradually changing reward value. We had measured licking of the monkeys during the recording of 12 of the 19 dopamine neurons with a tonic increase in activity in the value-increase condition (new Figure 2—figure supplement 6A) and during the recording of the 15 dopamine neurons with a tonic decrease in activity in the value-decrease condition (new Figure 2—figure supplement 6C). We analyzed these licking data to respond to the reviewer’s concern. First, we observed that the monkeys did not increase or decrease anticipatory licking as the reward value gradually changed. We did not expect this result because anticipatory licking correlated with reward value has been generally observed in previous studies. But, this result does not indicate that the monkeys did not predict the reward value from the CSs. Indeed, in the choice task in which two of the CSs were presented as options (Figure 1E), the monkeys chose the option associated with a larger reward (Figure 1F).

Although we observed that the monkeys did not increase or decrease licking as the reward value gradually changed, we analyzed the relationship between dopamine neuron activity and licking (new Figure 2—figure supplement 6B and D). We found that the activity of dopamine neurons was not influenced by licking. We added the description of these analysis and result in the revised manuscript as follows:

“Recent studies using rodents have shown that a subgroup of dopamine neurons increases their activity when animals simply initiate a body movement (da Silva et al., 2018; Howe and Dombeck, 2016), although such movement-related dopamine neuron activation has not been reported in primates. […] These dopamine neurons did not exhibit clear activity modulation around the onset (see Figure 2—figure supplement 6B for the dopamine neurons with a tonic increase in activity in the value-increase condition, and see Figure 2—figure supplement 6D for the dopamine neurons with a tonic decrease in activity in the value-decrease condition), suggesting that the tonic increase and decrease in dopamine neuron activity were not caused by licking.”

The reviewer further raised the possibility that the tonic increase and decrease in dopamine neuron activity might be caused by gaze. We therefore analyzed the relationship between dopamine neuron activity and eye movements, and found that the activity of dopamine neurons was not influenced by them (new Figure 2—figure supplement 7). We added the description of these analysis and result in the revised manuscript as follows:

“We also observed that the monkeys often made eye movements along the vertical bar stimulus (i.e., CS) while the reward value gradually changed (see Figure 2—figure supplement 7A and E for example trials in the value-increase and value-decrease conditions, respectively). […] These results suggest that the tonic increase and decrease in dopamine neuron activity were not caused by eye movements.”

5) Did the total amount of reward differ between blocks of trials? If so, was this reflected in the firing?

This is an important information. The total amount of reward was the same among blocks (10 ml). We added this information in the revised manuscript as follows:

“The total amount of reward was the same (10 ml) among blocks.”

6) Did tonic firing patterns change across blocks of trials and were they consistent between the first and second runs of the same trial block?

This is also an important point. To test whether the tonic activity increase and decrease patterns changed across blocks of trials, we compared the activity patterns between the first and later blocks (new Figure 2—figure supplement 5A and B). Consequently, we observed no significant change in the patterns. We added the description of these analysis and result in the revised manuscript as follows:

“Although each condition consisted of a block of 50 trials and was repeated once or more for each recording session (Figure 1D), the tonic increase and decrease in activity were consistently observed between the first and later blocks of each condition (Figure 2—figure supplement 5A, B). No neuron exhibited a significantly different regression slope between the first and later blocks (p > 0.05; comparison of two regression slopes).”

7) Since this was a blocked paradigm, some of the effect might be due to drift in the signal or loss of cells or emergence of new cells. How stable were the recording over time?

We observed that a subgroup of dopamine neurons exhibited a tonic activity increase in the value-increase condition, tonic activity decrease in the value-decrease condition, and phasic responses in the value-fixed condition. As the reviewer pointed out, these different activity patterns might be due to the loss of spikes or emergence of spikes of other neurons during single-unit recording. To respond to the reviewer’s concern, we compared the baseline firing rate of the dopamine neurons exhibiting the tonic activity increase (19 neurons) and decrease (15 neurons) among the three conditions. At the single neuron level, only a few of these neurons exhibited a significant difference in baseline firing rate among the three conditions. As a population, there was no significant difference in baseline firing rate among the conditions. These results suggest that the three different activity patterns were not due to the loss of spikes or emergence of other spikes during single-unit recording. We added the description of these analysis and result in the revised manuscript as follows:

“Furthermore, to exclude the possibility that the difference in activity patterns among the value-increase, value-decrease, and value-fixed conditions (i.e., tonic increase, tonic decrease, and phasic changes, respectively) was due to the loss of spikes or emergence of spikes of other neurons during single-unit recording, we compared the baseline firing rate (i.e., firing rate from 500 to 0 ms before the fixation point onset) of the 19 and 15 dopamine neurons among the three conditions. […] These results suggest that the three different activity patterns were not due to the loss of spikes or emergence of other spikes during single-unit recording.”

8) The authors might be going one step too far by suggesting this might be involved in action selection. Why didn't they record during the choice task?

We totally agree that our results are not sufficient to discuss the role of the tonic dopamine activity in action selection. We therefore removed the term “action selection” from the manuscript (the previous Abstract and Discussion had included this term). Instead, we discussed its role in regulating “motivational vigor of actions”, which can be expected from previous studies. The following are from the new Abstract and Discussion:

“Animal behavior is regulated based on the values of future rewards. […] Our findings suggest that dopamine neurons change their firing mode to effectively signal reward values in a given situation.”

“While the phasic dopamine activity, which is presumed to represent reward prediction errors, has been considered to provide teaching signals in reinforcement learning (Doya, 2002; Montague et al., 1996; Schultz et al., 1997), the tonic dopamine activity has been proposed to regulate motivation (Cagniard et al., 2006; Niv, 2007). […] That is, by tracking the changing values of future rewards, the tonic activity might enhance and suppress the motivational vigor of actions to obtain the rewards on a moment-by-moment basis.”